# SemiFL: Semi-Supervised Federated Learning for Unlabeled Clients with Alternate Training

**Enmao Diao**
Department of Electrical and Computer Engineering
Duke University
Durham, NC 27705, USA
`enmao.diao@duke.edu`

**Jie Ding**
School of Statistics
University of Minnesota-Twin Cities
Minneapolis, MN 55455, USA
`dingj@umn.edu`

**Vahid Tarokh**
Department of Electrical and Computer Engineering
Duke University
Durham, NC 27705, USA
`vahid.tarokh@duke.edu`

## Abstract

Federated Learning allows the training of machine learning models by using the computation and private data resources of many distributed clients. Most existing results on Federated Learning (FL) assume the clients have ground-truth labels. However, in many practical scenarios, clients may be unable to label task-specific data due to a lack of expertise or resource. We propose SemiFL to address the problem of combining communication-efficient FL such as FedAvg with Semi-Supervised Learning (SSL). In SemiFL, clients have completely unlabeled data and can train multiple local epochs to reduce communication costs, while the server has a small amount of labeled data. We provide a theoretical understanding of the success of data augmentation-based SSL methods to illustrate the bottleneck of a vanilla combination of communication-efficient FL with SSL. To address this issue, we propose alternate training to 'fine-tune global model with labeled data' and 'generate pseudo-labels with the global model.' We conduct extensive experiments and demonstrate that our approach significantly improves the performance of a labeled server with unlabeled clients training with multiple local epochs. Moreover, our method outperforms many existing SSFL baselines and performs competitively with the state-of-the-art FL and SSL results. Our code is available here.

## 1 Introduction

For billions of users around the world, mobile devices and Internet of Things (IoT) devices are becoming common computing platforms [1]. These devices produce a large amount of data that can be used to improve a variety of existing applications [2]. Consequently, it has become increasingly appealing to process data and train models locally from privacy and economic standpoints. To address this, distributed machine learning framework of Federated Learning (FL) has been proposed [3, 4]. This method aggregates locally trained model parameters in order to produce a global inference model without sharing private local data.

Most existing works of FL focus on supervised learning tasks assuming that clients have ground-truth labels. However, in many practical scenarios, most clients may not be experts in the task of interest to label their data. In particular, the private data of each client may be completely unlabeled. For instance, a healthcare system may involve a central hub ("server") with domain experts and a

36th Conference on Neural Information Processing Systems (NeurIPS 2022).

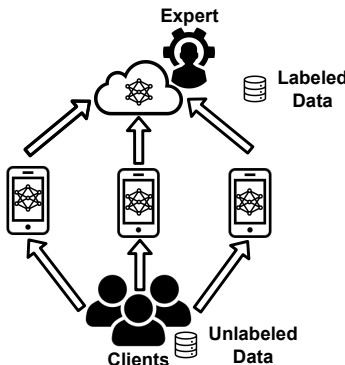

Figure 1: A resourceful server with labeled data can significantly improve its learning performance by working with distributed clients with unlabeled data without data sharing.

limited number of labeled data (such as medical records), together with many rural branches with non-experts and a massive number of unlabeled data. As another example, an autonomous driving startup ("server") may only afford beta-users assistance in labeling a road condition but desires to improve its modeling quality with the information provided by many decentralized vehicles that are not beta-users. The above scenarios naturally lead to the following important question: *How a server that hosts a labeled dataset can leverage clients with unlabeled data for a supervised learning task in the Federated Learning setting?*

We propose a new Federated Learning framework SemiFL to address the problem of Semi-Supervised Federated Learning (SSFL) as illustrated in Figure 1. We discover that it is challenging to directly combine the state-of-the-art SSL methods with the communication efficient federated learning methods such as FedAvg to allow local clients to train multiple epochs [4]. The key ingredient that enables SemiFL to allow unlabeled clients to train multiple local epochs is that we alternate the training of a labeled server and unlabeled clients to ensure that the quality of pseudo-labeling is highly maintained during training. In particular, we fine-tune the global model with labeled data and generate pseudo-labels only with the global model. We perform extensive empirical experiments to evaluate and compare our method with various baselines and state-of-the-art techniques. The results demonstrate that our method can outperform existing SSFL methods and perform close to the state-of-the-art of FL and SSL results. In particular, we contribute the following.

- We propose SemiFL in which clients have completely unlabeled data and can train multiple local epochs to reduce communication costs, while the server has a small amount of labeled data. We identify the difficulty of combining communication efficient FL method FedAvg [4] with the state-of-the-art SSL methods.

- We develop a theoretical analysis on strong data augmentation for SSL methods, the first in the literature to our best knowledge. We provide a theoretical understanding of the success of data augmentation-based SSL methods to illustrate the bottleneck of a vanilla combination of communication-efficient FL with SSL.

- To the best of our knowledge, we propose the first communication efficient SSFL method alternate training that can improve the performance of a labeled server by allowing unlabeled clients to train multiple local epochs, i.e., from $42\%$ to $88\%$ with $250$ labeled data, and from $77\%$ to $93\%$ accuracy with $4000$ labeled data on the CIFAR10 dataset. Moreover, our proposed method achieves $30\%$ improvement over the existing SSFL methods. Furthermore, SemiFL performs competitively with the state-of-the-art FL methods and SSL methods. i.e., only $1\%$ and $2\%$ away from the state-of-the-art FL and SSL results, respectively, for $4000$ labeled data on the CIFAR10 dataset.

The outline of the paper is given below. In Section 2, we review the related work. In Section 3, we identify the problem of combining SSL with communication-efficient FL methods, develop a theoretical analysis of how strong data augmentation can significantly improve the classification accuracy, and present the proposed SemiFL method with some intuitive explanations. In Section 4, we evaluate the empirical performance of the SemiFL. We make some concluding remarks in Section 5.

## 2   Related Work

**Federated Learning**   The goal of Federated Learning is to scale and speed up the training of distributed models [5, 6]. FedAvg [4] allows local clients to train multiple epochs to facilitate convergence. FedProx (Li et al., 2018) performs proximal regularization against global weights. FL counterparts of Batch Normalization [7–9] are developed to further enhance the performance. The use of local momentum and global momentum [10] have been shown to facilitate faster convergence. FedOpt [11] proposes federated versions of adaptive optimizers to improve performance over FedAvg.

**Semi-Supervised Learning**   Semi-Supervised Learning (SSL) refers to the general problem of learning with partially labeled data, especially when the amount of unlabeled data is much larger than that of the labeled data [12, 13]. The idea of self-training (namely to obtain artificial labels for unlabeled data from a pre-trained model) can be traced back to decades ago [14, 15]. Pseudo-labeling [16], a component of many recent SSL techniques [17], is a form of entropy minimization [18] by converting model predictions into hard labels. Consistency regularization [19] refers to training models via minimizing the distance among stochastic outputs [13, 19]. A theoretical analysis of consistency regularization was recently developed in [20]. More recently, It has been demonstrated that the technique of strong data augmentation can lead to better outcomes [21–24]. Strongly augmented examples are frequently found outside of the training data distribution, which has been shown to benefit SSL [25].

**Semi-Supervised Federated Learning (SSFL)**   Most existing FL works focus on supervised learning tasks, with clients having ground-truth labels. However, in many real-world scenarios, most clients are unlikely to be experts in the task of interest, an issue raised in a recent survey paper [26]. In the research line of SSFL, the work [27] splits model parameters for labeled server and unlabeled clients separately. Another related work [28] trains and aggregates the model parameters of the labeled server and unlabeled clients in parallel with group-wise reweights. Applications of SSFL to specific applications can be found in, e.g., [29, 30].

## 3   Method

### 3.1   Problem

In a supervised learning classification task, we are given a dataset $\mathcal{D} = \{x_i, y_i\}_{i=1}^{N}$, where $x_i$ is a feature vector, $y_i$ is an one-hot vector representing the class label in a $K$-class classification problem, and $N$ is the number of training examples. In a Semi-Supervised Learning classification task, we have two datasets, namely a supervised dataset $\mathcal{S}$ and an unsupervised dataset $\mathcal{U}$. Let $\mathcal{S} = \{x_s^i, y_s^i\}_{i=1}^{N_\mathcal{S}}$ be a set of $N_\mathcal{S}$ labeled data observations, and $\mathcal{U} = \{x_u^i\}_{i=1}^{N_\mathcal{U}}$ be a set of $N_\mathcal{U}$ unlabeled observations (without the corresponding true label $y_u^i$). It is often interesting to study the case where $N_\mathcal{S} \ll N_\mathcal{U}$.

In this work, we focus on Semi-Supervised Federated Learning (SSFL) with unlabeled clients, as illustrated in Figure 1. Assume $M$ clients and let $x_{u,m}$ denote the set of unsupervised data available at client $m = 1, 2, \cdots, M$. Similarly, let $(x_s, y_s)$ denote the set of labeled data available at the server. The server model is parameterized by model parameters $W_s$. The client models are parameterized respectively by model parameters $\{W_{u,1}, \ldots, W_{u,M}\}$. We assume that all models share the same model architecture, denoted by $f : (x, w) \mapsto f(x, w)$, which maps an input $x$ and parameters $W$ to a vector on the $K$-dimensional simplex, e.g., using softmax function applied to model outputs.

**Communication Efficient FL with SSL**   In the standard communication efficient FL scenario where clients can train multiple local epochs before model aggregation (i.e., FedAvg [4]), existing SSFL methods have difficulty in performing close to the state-of-the-art centralized SSL methods [27, 28, 31]. In fact, we will demonstrate in Table 1 that existing SSFL methods cannot outperform the case of training with only labeled data. This is somewhat surprising given that their underlying methods of training unlabeled data are similar.

As shown in Figure 2, SSL methods such as FixMatch can only work with FedSGD, which requires batch-wise gradient aggregation and thus is not communication efficient. This is because SSL methods, such as FixMatch and MixMatch, sample from both labeled and unlabeled datasets for every batch of training data with a carefully tuned ratio [23, 32]. Thus, it is not straightforward how we can combine the SSL method in a communication-efficient FL scenario where we train multiple local epochs. To understand the bottleneck of this vanilla combination, we need to understand better

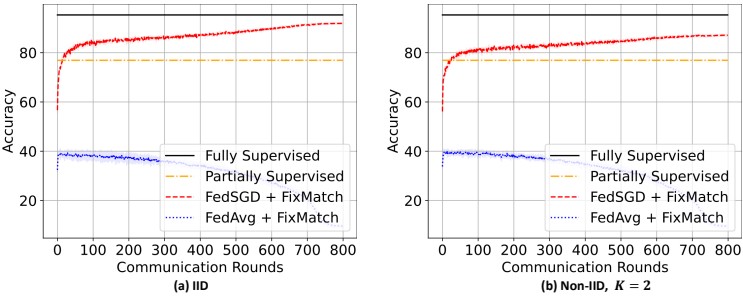

Figure 2: Results of CIFAR10 dataset with (a) IID and (b) Non-IID, $K = 2$ data partition and $N_{\mathcal{S}} = 4000$ with a vanilla combination of communication efficient FL with SSFL methods. The "Fully Supervised" and "Partially Supervised" refer to training a centralized model with full and $4000$ labeled data respectively.

why the state-of-the-art centralized SSL methods work. In section 3.2, we analyze the strong data augmentation for SSL and demonstrate that the success of FixMatch is due to using data augmentation on pseudo-labeled data of high quality.

## 3.2 Theoretical Analysis of Strong Data augmentation for SSL

Pseudo-labeling is widely used for labeling unlabeled data in SSL methods [16, 22, 32]. However, the quality (accuracy) of those pseudo-labels can be low, especially at the beginning of the training. In this light, several papers [22, 32] propose to hard-threshold or sharpen the pseudo-labels to improve the quantity of accurately labeled pseudo-labels. The problem with hard thresholding is that the data samples satisfying the confidence threshold have a small training loss. Therefore, the model cannot be significantly improved as it already performs well on the data above the threshold. To address this issue, we will use strong data augmentation [25, 32] to generate data samples that have larger training loss. The main idea is to construct a pseudo-labeling mechanism whereby our SSL method can generate more and more high-quality pseudo-labels during training. Meanwhile, the augmented data for model training can produce a more considerable drop in training loss than the original data.

To provide further insights into SSL, we develop a theoretical analysis of the strong data augmentation, which is a critical component of the state-of-the-art SSL method FixMatch [32] and SemiFL, and can be interesting in its own right. Intuitively, strong augmentation is a process that maps a data point (e.g., an image) from high quality to relatively low grade unilaterally. The low-quality data and their high-confidence pseudo-labels are then used for training so that there are sufficient "observations" in the data regime insufficiently covered by labeled data.

Our theory is based on an intuitive "adequate transmission" assumption, which means that the distribution of augmented data from high-confidence unlabeled data can adequately cover the data regime of interest during prediction. Consequently, reliable information exhibited from unlabeled data can be "transmitted" to data regimes that may have been insufficiently trained with labeled data, as illustrated in Figure 3. Instead of studying SSL in full generality, we restrict our attention to a class of nonparametric kernel-based classification learning [33–35] and derive analytically tractable statistical risk-rate analysis. More detailed background and technical details are included in the Appendix. We provide a simplified statement as follows.

**Theorem 1** *Under suitable assumptions, an SSL classifier $\hat{C}^{ssl}$ trained from $n_u$ unlabeled data and the strong data augmentation technique has a statistical risk bound at the order of $\mathcal{R}(\hat{C}^{ssl}) \sim n_u^{-q(\alpha+1)/\{q(\alpha+3+\rho)+d\}}$ where $d$, $q$, $\alpha$, $\rho$ are constants that describe the data dimension, smoothness of the conditional distribution function $(Y \mid X)$, class separability (or task difficulty), and inadequacy of transmission, respectively. The smaller $\rho$, the better risk bound. Moreover, suppose that $\hat{C}^l$ is the classifier trained from $n_l$ labeled data, where $n_l \sim n_u^\zeta$, $\zeta \in (0, 1)$. It can be verified that the bound of $\mathcal{R}(\hat{C}^u)$ is much smaller than that of $\mathcal{R}(\hat{C}^l)$ when $\zeta < \frac{q(\alpha+3)+d}{q(\alpha+3+\rho)+d}$. This provides an insight into the critical region of $n_u$ where significant improvement can be made from unlabeled data.*

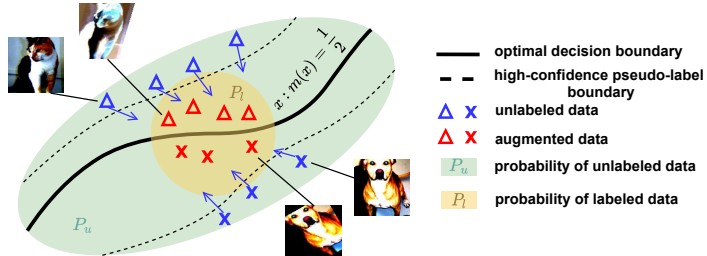

Figure 3: Illustration of the strong data augmentation-based SSL. We pick up an unlabeled point $(X \sim \mathbb{P}_u)$ with a high-confidence pseudo-label, obtain its hard-thresholded label ($\hat{Y}$, which is believed to be close to the ground truth), maneuver $X$ into $\tilde{X}$ (which is believed to represent the test distribution $\mathbb{P}_l$ to some extent), and then treat $(\hat{Y}, \tilde{X})$ as labeled data for training. Consequently, reliable task-specific information exhibited from unlabeled data can be transmitted to data regimes that may have been insufficiently trained with labeled data. Note that $\mathbb{P}_l$ denotes the labeled data distribution as well as the out-sample test data distribution (used to evaluate the learning performance). The above ideas are theoretically formalized in Subsection 3.2 and Appendix D.

## 3.3 Alternate Training

As depicted in Figure 4(a), existing SSFL works follow the state-of-the-art SSL methods to synchronize the training of supervised and unsupervised data [23, 32]. For example, FedMatch [27] and FedRGD [28] adopt a vanilla combination of FedAvg and FixMatch. They aggregate the server model trained from labeled data and clients' models trained from unlabeled data at each communication round in parallel and generate pseudo-labels for each batch of unlabeled data with the local training model. However, existing results [27, 28] indicate that this vanilla combination has difficulty performing close to the state-of-the-art SSL methods, even if the unlabeled clients are trained with the same SSL methods.

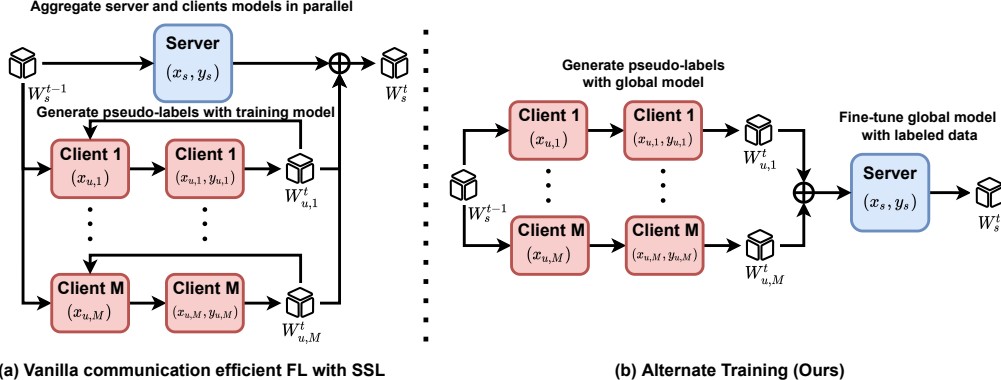

Figure 4: An illustration of (a) vanilla combination of communication efficient FL and SSL, and (b) Alternate Training (Ours). (a) The vanilla combination trains and aggregates server and client models in parallel and generates pseudo-labels with the training models for every batch of unlabeled data. (b) Alternate Training fine-tunes the aggregated global model with labeled data and generates pseudo-labels only once upon receiving the global model from the server.

In the communication efficient FL setting, we cannot guarantee an increase in the quality of pseudo-labels during training because we allow local clients to train multiple epochs, potentially deteriorating the performance (see Figure 2). Furthermore, the aggregation of a server model trained with ground-truth labels and a subset of client models trained with pseudo-labels does not constantly improve the performance of the global model over the previous communication round. A poorly aggregated global model of the last round produces worse-quality pseudo-labels. Subsequently, the performance of the aggregated model degrades in the next round.

To maintain and improve the quality of our generated pseudo-labels during training, we propose to train the labeled server and unlabeled clients in an alternate manner, as illustrated in Figure 4(b). In particular, our approach consists of two important components:

• **Fine-tune global model with labeled data** At each round, the server will retrain the global model with the labeled data. In this way, the server can provide a comparable or better model than the previous round for the active clients in the next round to generate pseudo-labels. On the contrary, the vanilla method aggregates server and client models in parallel. As a result, the quality of generated pseudo-labels will gradually degrade and thus deteriorate the performance.

• **Generate pseudo-labels with global model** We will label the unlabeled data once the active clients immediately receive the global model from the server. This way, pseudo-labels' quality will not degrade during the local training. On the contrary, the vanilla method labels every batch of data during the training of unlabeled clients. As a result, the quality of generated pseudo-labels will gradually degrade during local training, thus deteriorating performance.

Our proposed approach ensures that the clients can continually generate better quality pseudo-labels during training. Our experimental studies show that the proposed method can significantly improve the performance of the labeled server and performs competitively even with the state-of-the-art FL and centralized SSL methods. The limitation of our approach is that we need to update the aggregated client model with labeled data from the server, which will delay the computation time. We will conduct an ablation study on each component of alternative training in Table 2.

### 3.4 The SemiFL Algorithm

We summarize the pseudo-code of the proposed solution in Algorithm 1. At each iteration $t$, the server will first update the model with the standard supervised loss $L_s$ for local epochs $E$ with data batch $(x_b, y_b)$ of size $B_s$ randomly split from the supervised dataset $\mathcal{D}_s$, using

$$L_s = \ell(f(\alpha(x_b), W_s), y_b), \quad W_s = W_s - \eta \nabla_W L_s, \tag{1}$$

where $\alpha(\cdot)$ represents a weak data augmentation, such as random horizontal flipping and random cropping, that maps one image to another. Subsequently, the server updates the static Batch Normalization (sBN) statistics [9] (which is discussed in Appendix B). Next, the server distributes server model parameters $W_s$ to a subset of clients. We denote the proportion of active clients at each communication round $t$ as activity rate $C_t \in (0, 1]$. Without loss of generality, we assume that $C_t = C$ is a constant over time. After each active local client, say client $m$, receives the transmitted $W_s$, it generates pseudo-labels $y_{u,m}$ as follows:

$$W_{u,m} \leftarrow W_s, \quad y_{u,m} = f(\alpha(x_{u,m}), W_{u,m}). \tag{2}$$

Each local client will construct a high-confidence dataset $\mathcal{D}_{u,m}^{\text{fix}}$ inspired by FixMatch [32] at each iteration $t$, defined as:

$$\mathcal{D}_{u,m}^{\text{fix}} = \{(x_{u,m}, y_{u,m}) \text{ with } \max(y_{u,m}) \geq \tau\}. \tag{3}$$

for a global confidence threshold $0 < \tau < 1$ pre-selected by all clients. If for some client $m$, we have $\mathcal{D}_{u,m}^{\text{fix}} = \emptyset$ then it will stop and refrain from transmission to the server. Otherwise, we will sample with replacement to construct a dataset inspired by MixMatch [23]. In other words,

$$\mathcal{D}_{u,m}^{\text{mix}} = \texttt{Sample } |\mathcal{D}_{u,m}^{\text{fix}}| \texttt{ with replacement}\{(x_{u,m}, y_{u,m})\}, \tag{4}$$

where $|\mathcal{D}_{u,m}^{\text{fix}}|$ denotes the number of elements of $\mathcal{D}_{u,m}^{\text{fix}}$. Thus $|\mathcal{D}_{u,m}^{\text{mix}}| = |\mathcal{D}_{u,m}^{\text{fix}}|$. Subsequently, client $m$ trains its local model for $E$ epoch to speed up convergence [4]. For each local training epoch of the client $m$, it randomly splits local data $\mathcal{D}_{u,m}^{\text{fix}}, \mathcal{D}_{u,m}^{\text{mix}}$ into batches $\mathcal{B}_{u,m}^{\text{fix}}, \mathcal{B}_{u,m}^{\text{mix}}$ of size $B_m$. For each batch iteration, as in [36], client $m$ constructs Mixup data from one particular data batch $(x_b^{\text{fix}}, y_b^{\text{fix}}), (x_b^{\text{mix}}, y_b^{\text{mix}})$ by

$$\lambda_{\text{mix}} \sim \text{Beta}(a, a), \quad x_{\text{mix}} \leftarrow \lambda_{\text{mix}} x_b^{\text{fix}} + (1 - \lambda_{\text{mix}}) x_b^{\text{mix}},$$

where $a$ is the Mixup hyperparameter. Next, client $m$ defines the "fix" loss $L_{\text{fix}}$ [32] and "mix" loss $L_{\text{mix}}$ [24] by

$$L_{\text{fix}} = \ell(f(\mathcal{A}(x_b^{\text{fix}}), W_{u,m}), y_b^{\text{fix}}),$$
$$L_{\text{mix}} = \lambda_{\text{mix}} \cdot \ell(f(\alpha(x_{\text{mix}}), W_{u,m}), y_b^{\text{fix}}) + (1 - \lambda_{\text{mix}}) \cdot \ell(f(\alpha(x_{\text{mix}}), W_{u,m}), y_b^{\text{mix}})). \tag{5}$$

**Algorithm 1** Semi-Supervised Federated Learning with Alternate Training for Unlabeled Clients

---

**Input:** Unlabeled data $x_{u,1:M}$ distributed on $M$ local clients, activity rate $C$, the number of communication rounds $T$, the number of local training epochs $E$, server and client respective batch sizes $B_s$ and $B_m$, local learning rate $\eta$, server model parameterized by $W_s$ client models parameterized by $\{W_{u,1}, \ldots, W_{u,M}\}$, weak data augmentation function $\alpha(\cdot)$, strong data augmentation function $\mathcal{A}(\cdot)$, confidence threshold $\tau$, Mixup hyper-parameter $a$, loss hyperparameter $\lambda$, common model architecture function $f(\cdot)$

**System executes:**

  **for** each communication round $t = 1, 2, \ldots T$ **do**

    $W_s^t \leftarrow$ **ServerUpdate**$(x_s, y_s, W_s^t)$

    Update the sBN statistics

    $S_t \leftarrow \max(\lfloor C \cdot M \rfloor, 1)$ active clients uniformly sampled without replacement

    **for** each client $m \in S_t$ **in parallel do**

      Distribute server model parameters to local client $m$, namely $W_{u,m}^t \leftarrow W_s^t$

      $W_{u,m}^t \leftarrow$ **ClientUpdate**$(x_{u,m}, W_{u,m}^t)$

    **end**

    Receive model parameters from $M_t$ clients, and calculate $W_s^t = M_t^{-1} \sum_{m=1}^{M_t} W_{u,m}^t$

  **end**

  $W_s^T \leftarrow$ **ServerUpdate**$(x_s, y_s, W_s^T)$

  Update the sBN statistics

**ServerUpdate** $(x_s, y_s, W_s)$**:**

  Construct supervised dataset $\mathcal{D}_s = (x_s, y_s)$

  **for** each local epoch $e$ from 1 to $E$ **do**

    $\mathcal{B}_s \leftarrow$ Randomly split local data $\mathcal{D}_s$ into batches of size $B_s$

    **for** batch $(x_b, y_b) \in \mathcal{B}_s$ **do**

      $L_s \leftarrow \ell(f(\alpha(x_b), W_s), y_b)$

      $W_s \leftarrow W_s - \eta \nabla_W L_s$

    **end**

  **end**

  Return $W_s$

**ClientUpdate** $(x_{u,m}, W_{u,m})$**:**

  Generate pseudo-labels with weakly augmented data $\alpha(x_{u,m})$, namely $y_{u,m} = f(\alpha(x_{u,m}), W_{u,m})$

  Construct FixMatch dataset, namely $\mathcal{D}_{u,m}^{\text{fix}} = \{(x_{u,m}, y_{u,m}) \ \texttt{with} \ \max(y_{u,m}) \geq \tau\}$

  If $\mathcal{D}_{u,m}^{\text{fix}} = \emptyset$ then Stop. Return.

  Construct an equal-size Mixup dataset, namely

  $\mathcal{D}_{u,m}^{\text{mix}} = \ \texttt{Sample} \ |\mathcal{D}_{u,m}^{\text{fix}}| \ \texttt{with replacement}\{(x_{u,m}, y_{u,m})\}$

  **for** each local epoch $e$ from 1 to $E$ **do**

    $\mathcal{B}_{u,m}^{\text{fix}}, \mathcal{B}_{u,m}^{\text{mix}} \leftarrow$ Randomly split local data $\mathcal{D}_{u,m}^{\text{fix}}, \mathcal{D}_{u,m}^{\text{mix}}$ into batches of size $B_m^{\text{fix}}, B_m^{\text{mix}}$

    **for** batch $(x_b^{\text{fix}}, y_b^{\text{fix}}), (x_b^{\text{mix}}, y_b^{\text{mix}}) \in \mathcal{B}_{u,m}^{\text{fix}}, \mathcal{B}_{u,m}^{\text{mix}}$ **do**

      $\lambda_{\text{mix}} \sim \text{Beta}(a, a)$

      $x_{\text{mix}} \leftarrow \lambda_{\text{mix}} x_b^{\text{fix}} + (1 - \lambda_{\text{mix}}) x_b^{\text{mix}}$

      $L_{\text{fix}} \leftarrow \ell(f(\mathcal{A}(x_b^{\text{fix}}), W_{u,m}), y_b^{\text{fix}})$

      $L_{\text{mix}} \leftarrow \lambda_{\text{mix}} \cdot \ell(f(\alpha(x_{\text{mix}}), W_{u,m}), y_b^{\text{fix}}) + (1 - \lambda_{\text{mix}}) \cdot \ell(f(\alpha(x_{\text{mix}}), W_{u,m}), y_b^{\text{mix}}))$

      $W_{u,m} \leftarrow W_{u,m} - \eta \nabla_W (L_{\text{fix}} + \lambda \cdot L_{\text{mix}})$

    **end**

  **end**

  Return $W_{u,m}$ and send it to the server

---

Here, $\mathcal{A}$ represents a strong data augmentation mapping, e.g., the RandAugment [37] used in our experiments, and $\ell$ is often the cross entropy loss for classification tasks. Finally, client $m$ performs a gradient descent step with

$$W_{u,m} = W_{u,m} - \eta \nabla_W (L_{\text{fix}} + \lambda \cdot L_{\text{mix}}), \tag{6}$$

where $\lambda > 0$ is a hyperparameter set to be one in our experiments. After training for $E$ local epochs, client $m$ transmits $W_{u,m}$ to the server.

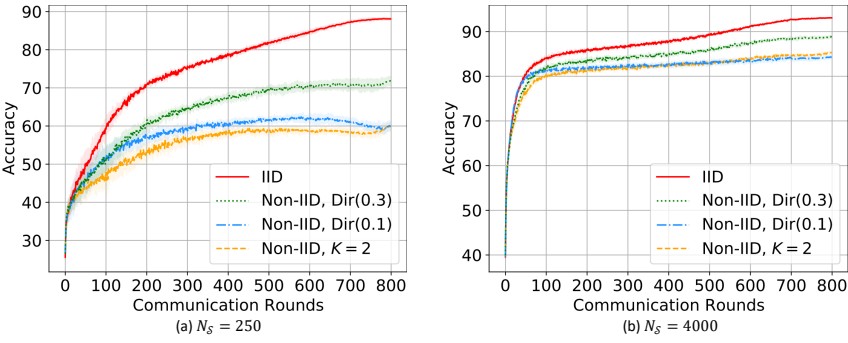

Figure 5: Results of CIFAR10 dataset with (a) $N_{\mathcal{S}} = 250$ and (b) $N_{\mathcal{S}} = 4000$.

Without loss of generality, assume that clients $1, 2, \cdots, M_t$ have sent their models to the server at time $t$. The server then aggregates client model parameters $\{W_{u,1}, \ldots, W_{u,M_t}\}$ by $W_s = M_t^{-1} \sum_{m=1}^{M_t} W_{u,m}$ [4]. This process is then repeated for multiple communication rounds $T$. After the training is finished, the server will further fine-tune the aggregated global model by additional training with the server's supervised data using its supervised loss $L_s$. Finally, it will update the sBN statistics one final time.

## 4    Experiments

**Experimental setup**  To evaluate our proposed method, we conduct experiments with CIFAR10, SVHN, and CIFAR100 datasets [38, 39]. To compare our method with existing FL and SSFL methods, we follow the standard communication efficient FL setting, which was originally used in FedAvg [4] and widely adopted by following works, such as [9, 40, 41]. We have 100 clients throughout our experiments, and the activity rate per communication round is $C = 0.1$. We uniformly assign the same number of data examples for IID data partition to each client. For a balanced Non-IID data partition, we ensure each client has data at most from $K$ classes and the sample size of each class is the same. We set $K = 2$ because it is the most label-skewed case for classification, and it has been evaluated in [9, 40, 41]. For unbalanced Non-IID data partition, we sample data for each client from a Dirichlet distribution $\text{Dir}(\alpha)$ [41, 42]. As $\alpha \to \infty$, it reduces to IID data partition. We perform experiments with $\alpha = \{0.1, 0.3\}$.

To compare our method with the state-of-the-art SSL methods, we follow the experimental setup in [32]. We use Wide ResNet28x2 [43] as our backbone model for CIFAR10 and SVHN datasets and Wide ResNet28x8 for CIFAR100 datasets throughout our experiments. The number of labeled data at the server for CIFAR10, SVHN, and CIFAR100 datasets $N_{\mathcal{S}}$ are $\{250, 4000\}$, $\{100, 2500\}$, and $\{2500, 10000\}$ respectively. We conduct four random experiments for all the datasets with different seeds, and the standard errors are shown inside the parentheses for tables and by error bars in the figures. We demonstrate our experimental results in Table 1 and the learning curves of CIFAR10, SVNH, and CIFAR100 datasets in Figure 5, 7, and 8. Further details are included in the Appendix.

**Comparison with SSL methods**  We demonstrate the results of Fully Supervised and Partially Supervised cases and existing SSL methods for comparison in Table 1. The Fully Supervised case refers to all data being labeled, while in the Partially Supervised case, we only train the model with the partially labeled $N_{\mathcal{S}}$ data. Our results significantly outperform the Partially Supervised case. In other words, SemiFL can substantially improve the performance of a labeled server with unlabeled clients in a communication-efficient scenario. Our method performs competitively with the state-of-the-art SSL methods for IID data partition. Moreover, it is foreseeable that as the clients become more label-skewed for Non-IID data partition, the performance of our method degrades. However, even the most label-skewed unlabeled clients can improve the performance of the labeled server using our approach. A limitation of our work is that as the supervised data size decreases, the performance of SemiFL degrades more than the centralized SSL methods. We believe it is because we cannot train labeled and unlabeled data simultaneously in one data batch.

**Comparison with FL and SSFL methods**  We compare our results with the state-of-the-art FL and SSFL methods in Table 1. We demonstrate that SemiFL can perform competitively with the state-of-the-art FL result trained with fully supervised data. It is worth mentioning that SSFL may

Table 1: Comparison of SemiFL with the Baselines, SSL, FL, and SSFL methods. SemiFL improves the performance of the labeled server, SemiFL significantly outperforms the existing SSFL methods, and performs close to the state-of-the-art FL and SSL methods.

| | Dataset | CIFAR10 | | SVHN | | CIFAR100 | |
|---|---|---|---|---|---|---|---|
| | Number of Supervised | 250 | 4000 | 250 | 1000 | 2500 | 10000 |
| Baseline | Fully Supervised | 95.3(0.1) | | 97.3(0.0) | | 79.3(0.1) | |
| | Partially Supervised | 42.4(1.8) | 76.9(0.2) | 77.1(2.9) | 90.4(0.5) | 27.2(0.7) | 59.3(0.1) |
| | $\Pi$-Model [13] | 45.7(4.0) | 86.0(0.4) | 81.0(1.9) | 92.5(0.4) | 42.8(0.5) | 62.1(0.1) |
| | Pseudo-Labeling [44] | 50.2(0.4) | 83.9(0.3) | 79.8(1.1) | 90.1(0.6) | 42.6(0.5) | 63.8(0.2) |
| | Mean Teacher [44] | 67.7(2.3) | 90.8(0.2) | 96.4(0.1) | 96.6(0.1) | 46.1(0.6) | 64.2(0.2) |
| SSL | MixMatch [23] | 89.0(0.9) | 93.6(0.1) | 96.0(0.2) | 96.5(0.3) | 60.1(0.4) | 71.7(0.3) |
| | UDA [22] | 91.2(1.1) | 95.1(0.2) | 94.3(2.8) | 97.5(0.2) | 66.9(0.2) | 75.5(0.3) |
| | ReMixMatch [24] | 94.6(0.1) | 95.3(0.1) | 97.1(0.5) | 97.4(0.1) | 72.6(0.3) | 77.0(0.6) |
| | FixMatch [32] | 94.9(0.7) | 95.7(0.1) | 97.5(0.4) | 97.7(0.1) | 71.7(0.1) | 77.4(0.1) |
| Non-IID, $K = 2$ | FL   HeteroFL [9] | 51.5(3.6) | | 72.3(4.4) | | 3.1(0.3) | |
| | SSFL   FedMatch [27] | 41.3(1.1) | 58.3(1.0) | 58.2(3.1) | 84.3(1.0) | 17.7(0.5) | 30.5(0.8) |
| | FedRGD [28] | 32.7(3.6) | 48.9(1.4) | 21.2(2.2) | 21.6(2.3) | 13.8(1.4) | 26.5(3.0) |
| | SemiFL | **60.0(0.9)** | **85.3(0.3)** | **87.5(1.1)** | **92.2(0.8)** | **35.2(0.3)** | **62.1(0.4)** |
| Non-IID, $\mathrm{Dir}(0.1)$ | FL   HeteroFL [9] | 85.0(0.6) | | 95.8(0.1) | | 74.0(0.4) | |
| | SSFL   FedMatch [27] | 41.6(1.0) | 58.9(0.7) | 58.4(3.4) | 84.3(0.6) | 17.5(0.5) | 30.8(0.6) |
| | FedRGD [28] | 31.5(2.9) | 45.2(0.8) | 20.0(4.0) | 23.8(3.4) | 13.4(1.3) | 23.6(2.6) |
| | SemiFL | **63.0(0.6)** | **84.5(0.4)** | **91.2(0.3)** | **93.0(0.5)** | **49.0(1.0)** | **68.0(0.2)** |
| Non-IID, $\mathrm{Dir}(0.3)$ | FL   HeteroFL [9] | 91.6(0.1) | | 96.8(0.0) | | 76.9(0.1) | |
| | SSFL   FedMatch [27] | 41.2(1.1) | 58.4(0.6) | 59.1(2.8) | 84.0(1.1) | 17.8(0.4) | 31.1(0.5) |
| | FedRGD [28] | 32.5(3.0) | 46.9(1.6) | 24.8(5.1) | 22.0(3.9) | 13.1(2.0) | 23.8(1.9) |
| | SemiFL | **71.9(1.2)** | **88.9(0.3)** | **94.0(0.5)** | **95.2(0.2)** | **54.9(1.4)** | **70.0(0.3)** |
| IID | FL   HeteroFL [9] | 94.3(0.1) | | 97.5(0.0) | | 77.8(0.2) | |
| | SSFL   FedMatch [27] | 41.7(1.1) | 58.6(0.5) | 58.6(3.0) | 84.3(0.9) | 17.6(0.3) | 31.3(1.0) |
| | FedRGD [28] | 33.2(1.9) | 47.8(1.7) | 21.3(6.5) | 20.7(1.1) | 13.3(1.4) | 23.8(2.6) |
| | SemiFL | **88.2(0.3)** | **93.1(0.1)** | **96.8(0.3)** | **96.9(0.1)** | **61.3(1.2)** | **72.1(0.2)** |

outperform FL methods in the Non-IID data partition case because the server has a small set of labeled IID data. We also demonstrate that our method significantly outperforms existing SSFL methods. Existing SSFL methods fail to perform closely to the state-of-the-art centralized SSL methods, even if their underlying SSL methods are the same. Moreover, existing SSFL methods cannot outperform the Partially Supervised case, indicating that they deteriorate the performance of the labeled server. In particular, FedMatch allocates disjoint model parameters for the server and clients, and FedRGD assigns a higher weight for the server model for aggregation. Both methods do not directly fine-tune the global model with labeled data and generate pseudo-labels with the received global model. To our best knowledge, *the proposed SemiFL is the first SSFL method that actually improves the performance of the labeled server and performs close to the state-of-the-art FL and SSL methods.*

**Ablation studies** We conduct ablation studies on SemiFL and demonstrate the results in Table 2. Based on our extensive experiments, it is evident that "Fine-tune global model with labeled data" and "Generate pseudo-labels with global model" are the critical components of the proposed 'Alternate Training' method for the success of our method. We also conduct an ablation study on static Batch Normalization(sBN), the number of local training epochs, the Mixup data augmentation, and global SGD momentum. The detailed results can be found in the Appendix.

## 4.1 Quality of Pseudo Labeling

We measure the quality of Pseudo-Labeling for Semi-Supervised Learning from three aspects, including the accuracy of pseudo-labels (Pseudo Accuracy), the accuracy of thresholded pseudo-labels (Threshold Accuracy), and the ratio of pseudo-labeled data (Label Ratio) with CIFAR10 dataset in Figure 6. We perform ablation studies on our proposed method by measuring the quality of Pseudo-Labeling. The results demonstrate that our proposed alternative training, the combination of 'Fine-tune global model with labeled data' and 'Generate pseudo-labels with global model,' can produce pseudo labels of much better quality when clients have completely unlabeled data and train multiple local epochs.

Table 2: Ablation study on each component of alternative training with CIFAR10 dataset. The combination of "Fine-tune global model with labeled data" and "Generate pseudo-labels with global model" significantly improves the performance.

| Method | Fine-tune global model with labeled data | Generate pseudo-labels with global model | Accuracy | |
|---|---|---|---|---|
| | | | Non-IID, $K = 2$ | IID |
| Fully Supervised | N/A | | 95.33 | |
| Partially Supervised | | | 76.92 | |
| FedAvg + FixMatch | ✗ | ✗ | 41.01 | 40.26 |
| | ✗ | ✓ | 48.89 | 47.03 |
| SemiFL | ✓ | ✗ | 80.42 | 81.70 |
| | ✓ | ✓ | **85.34** | **93.10** |

Figure 6: Ablation studies of alternative training by measuring the quality of Pseudo Labeling with CIFAR10 dataset. 'Fine Tune' and 'Global' refer to our proposed method, 'Fine-tune global model with labeled data' and 'Generate pseudo-labels with global model,' respectively. 'Average' refers to the vanilla FL method, which directly takes the average of the model parameters of the labeled server and unlabeled clients. 'Training' refers to generating pseudo-labels at each batch of local training.

## 5 Conclusion

In this work, we propose a new communication-efficient Federated Learning (FL) framework named SemiFL to address the problem of Semi-Supervised Federated Learning (SSFL) for unlabeled clients. We identify the difficulty of combining communication-efficient Federated Learning (FL) with state-of-the-art Semi-Supervised Learning (SSL). We develop a theoretical analysis of strong data augmentation for SSL, which illustrates the bottleneck of vanilla combination. We propose to train the labeled server and unlabeled clients in an alternate manner by 'fine-tune global model with labeled data' and 'generate pseudo-labels with global model.' We utilize several training techniques and establish a strong benchmark for SSFL. Extensive experimental studies demonstrate that our communication-efficient method can significantly improve the performance of a labeled server with unlabeled clients. Moreover, we show that SemiFL can perform competitively with the state-of-the-art centralized SSL and fully supervised FL methods. Our study provides a practical SSFL framework that extends the scope of FL applications.

## Acknowledgments

The work of Enmao Diao and Vahid Tarokh was supported by the Office of Naval Research (ONR) under grant number N00014-18-1-2244. The work of Jie Ding was supported by the National Science Foundation (NSF) under grant number DMS-2134148.

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
