# OpenReview forum: "SemiFL: Semi-Supervised Federated Learning for Unlabeled Clients with Alternate Training"
_NeurIPS.cc/2022/Conference — NeurIPS 2022 Accept_

### Official Review · Reviewer_Y1ff · 2022-07-10

**Rating:** 3
**Confidence:** 4
**Soundness:** 2 fair
**Presentation:** 2 fair
**Contribution:** 2 fair

**Summary:**

This paper proposes SemiFL to address the problem of Semi-Supervised Federated Learning (SSFL). SemiFL alternates training of labelled server data and unlabeled clients to maintain pseudo-labeling quality during training, allowing unlabeled clients to train multiple local epochs. This method uses labelled data to fine-tune the global model and generate pseudo-labels.

**Questions:**

-	Please explain the novelty of SemiFL. Are there any technological challenges?
-	The proposed method SemiFL leverages the unlabeled data on local clients. However, in real applications, the data in local clients are private, so the server has no information about numbers and categories of data included in clients. And maybe the total amount of data is imbalanced or categories are not included in the server. How will SemiFL handle this situation? The authors should discuss these issues.
-	The proposed model of this paper mainly focused on the model level. How to deal with the possible in/consistency in the feature space?


**Limitations:**

-	The comparison of accuracy results of SSL methods is not essential, because the proposed method only use Fixmatch and Mixmatch in this paper. Some FSSL baselines are missing, like FedBYOL, FedU, etc. Please include the results of more recent baseline methods.
-	Please supplement theoretical analysis to support the model designed, e.g., convergence analysis.

**Strengths And Weaknesses:**

Strength
-	The proposed method is clear and easy to understand.
-	The paper is well-organized.

Weakness
-	The novelty of this paper is low, just a simple application of Fixmatch/Mixmatch in federated learning.
-	Missing important FSSL baselines, e.g., FedBYOL and FedU.
-	The theoretical proof given in this paper is only an analysis of data augmentation.

---

> ### Author Response · Authors · 2022-08-02
> **Response**
>
> Thank you for your time and constructive comments. We have addressed all the comments below. The following major changes will be included in the revision. We hope the responses and planned revisions will be viewed favorably.
>
> 1. Weakness - The novelty of this paper is low, just a simple application of Fixmatch/Mixmatch in federated learning. - Missing important FSSL baselines, e.g., FedBYOL and FedU. - The theoretical proof given in this paper is only an analysis of data augmentation.
> > We believe our method is not a simple application of Fixmatch in federated learning based on the following reasons. 1) From a methodological perspective, we propose alternative training, including `Fine-tune global model with labeled data' and `Generate pseudo-labels with global model'. To our best knowledge, these methods are new in the context of semi-supervised federated learning. Our proposed method is also simple and effective. In Table 2 and Section C.4 of the appendix (highlighted in the revision), we provided ablation studies of our proposed alternative training to demonstrate that our proposed method is the key to improving the performance of the labeled server. 2) FedBYOL is about self-supervised learning with federated learning, and FedU is about unsupervised learning with federated learning. Both methods are not in the context of semi-supervised learning with federated learning. Existing SSFL methods fail to improve the performance of the labeled server in communication efficient federated learning settings. 3) Furthermore, to our best knowledge, we propose the first theoretical analysis of strong data augmentation for semi-supervised learning.
> 2. Please explain the novelty of SemiFL. Are there any technological challenges?
> > The technological challenge is that existing Semi-Supervised Federated Learning (SSFL) methods cannot improve the performance of the labeled server in a communication efficient federated learning scenario, such as FedAvg, which allows local clients to train multiple epochs. In other words, existing methods do not leverage the unlabeled data at clients and perform worse than the case of training with only labeled data at the server. To our best knowledge, SemiFL is the first SSFL method that actually improves the performance of the labeled server and performs close to the state-of-the-art FL and SSL methods.
> 3. The proposed method SemiFL leverages the unlabeled data on local clients. However, in real applications, the data in local clients are private, so the server has no information about numbers and categories of data included in clients. And maybe the total amount of data is imbalanced or categories are not included in the server. How will SemiFL handle this situation? The authors should discuss these issues.
> > We perform ablation studies on the label skew heterogeneity in Table 1 where the data are split in a Non-IID fashion. We assume that the server maintains an IID dataset. This is also assumed by prior works, including FedMatch and FedRGD. In practice, if the labels at the server are imbalanced, one can adopt techniques like oversampling during fine-tuning to address such problems.
> 4. The proposed model of this paper mainly focused on the model level. How to deal with the possible in/consistency in the feature space?
> > Our paper focuses on the availability of labels rather than inconsistency in the feature space. In Table 1, we demonstrate that our method works well for FL with Non-IID data partition.
> 5. The comparison of accuracy results of SSL methods is not essential, because the proposed method only use Fixmatch and Mixmatch in this paper. Some FSSL baselines are missing, like FedBYOL, FedU, etc. Please include the results of more recent baseline methods.
> > FedBYOL is about self-supervised learning with federated learning, and FedU is about unsupervised learning with federated learning. Both methods are not in the context of semi-supervised learning with federated learning.

---

> > ### Author Response · Authors · 2022-08-02
> > **Response continue**
> >
> > 6. Please supplement theoretical analysis to support the model designed, e.g., convergence analysis.
> > > We agree that the paper would be stronger if some convergence results could be derived. However, we found it difficult to develop a comprehensive theory to understand the combination of FixMatch with Federated Learning. In fact, we are not aware of prior theoretical results on the strong data augmentation method used in FixMatch. Therefore, we focus on understanding why and when FixMatch may outperform conventional semi-supervised learning methods. In our original submission, we theoretically analyzed the effect of strong data augmentation on the semi-supervised learning method. In Section C.4 of the appendix (highlighted in the revision), we provided ablation studies of our proposed alternative training by measuring the quality of pseudo-labeling to demonstrate that thresholded pseudo-labels lead to higher accuracy. In summary, we found that 1) thresholded pseudo-labels have much higher accuracy than those without thresholding, and 2) strong data augmentation produces less confident predictions to create a larger loss drop compared with using the originally high confident data.

---

> ### Author Response · Authors · 2022-08-06
> **A kind reminder**
>
> Dear Reviewer Y1ff,
>
> We would like to thank you again for the time you dedicated to reviewing our paper and your valuable comments. We believe that we have addressed your concerns. Since the end of discussion period is getting close and we have not heard back from you yet, we would appreciate if you kindly let us know of any other concerns you may have, and if we can be of any further assistance in clarifying any other issues.
>
> Thanks a lot again, and with sincerest best wishes
>
> Authors

---

> ### Author Response · Authors · 2022-08-09
> **A kind reminder**
>
> Dear Reviewer Y1ff,
>
> We apologize for any inconvenience that our message may cause in advance. Again, we would like to thank you for the time you dedicated to reviewing our paper and your valuable comments. We believe that we have addressed your concerns. Since the end of discussion period is close and we have not heard back from you yet, we would appreciate if you kindly let us know of any concerns you may have, and if we can be of any further assistance in clarifying any other issues. We humbly remain at your disposal.
>
> Thanks a lot again, and with best wishes,
>
> Authors

---

### Official Review · Reviewer_sMTb · 2022-07-10

**Rating:** 5
**Confidence:** 4
**Soundness:** 2 fair
**Presentation:** 2 fair
**Contribution:** 2 fair

**Summary:**

This paper tackles a practical problem of federated learning where labeled data is completely unavailable at local clients whereas the global server has a small set of labeled data. This paper proposes the SemiFL method, which finetunes global models with labeled data while generating pseudo-labels with the global model at local training phase. This paper demonstrates the effectiveness of their proposed model compared to the existing SSFL, FL, and SSL methods.


**Questions:**

Please cite the prior works correctly (not arXiv). i.e. FedRGD is published at IEEE International Conference on Big Data (2021), FedMatch is published at International Conference on Learning Representations (2021).

Please clarify the description of prior works in 3.3 Alternate Training. For “Vanilla communication efficient FL with SSL” methods, it is correct that FedRGD [28] performas aggregation of server and client model in parallel as you described in Fig 4 (a). For FedMatch [27], however, as far as I know, it performs disjoint learning (a bit similar to the alternate training) which trains the global model with labeled data at server “after” local training with unlabeled data is done for the labels-at-server scenario. However, in line 157 to 161, it is not correctly described and so readers may be confused while reading the part.


**Limitations:**

No negative societal impact is expected.

**Strengths And Weaknesses:**

Clarity: (High) This paper is well-organized and easy-to-read. The authors smoothly develop their logic and key concepts throughout the paper. The figures are illustrated clearly.

Quality: (High) This paper focuses on the practical problem of FL. Its theoretical analysis is convincing. The experiments are carefully designed.

Originality: (Low) I think the critical weakness of this paper is the originality. Their proposed training algorithms seems to be 1) highly based on existing SSL methods  i.e, losses from FixMatch and Mixup, and 2) the alternate training is a bit simple to consider as a novel and significant approach.

Significance: (Moderate) The performance improvement compared to the existing SSFL methods seems significant. However, considering the originality of the proposed method, it is hard to say the overall significance of this paper is strong.

---

> ### Author Response · Authors · 2022-08-02
> **Response**
>
> Thank you for your time and constructive comments. We have addressed all the comments below. The following major changes will be included in the revision. We hope the responses and planned revisions will be viewed favorably.
>
> 1. Originality
> > We believe our contributions are novel in the following aspects. 1)  From a methodological perspective, we propose alternative training, including `Fine-tune global model with labeled data` and `Generate pseudo-labels with global model` are new in the context of semi-supervised federated learning. Our proposed method is also simple and effective. In Table 2 and Section C.4 of the appendix (highlighted in the revision), we provided ablation studies of our proposed alternative training to demonstrate that our proposed method is the key to improving the performance of the labeled server.
> 2. Please cite the prior works correctly (not arXiv). i.e. FedRGD is published at IEEE International Conference on Big Data (2021), FedMatch is published at International Conference on Learning Representations (2021).
> > We have corrected the citation as you suggested in the revision.
> 3. Please clarify the description of prior works in 3.3 Alternate Training. For “Vanilla communication efficient FL with SSL” methods, it is correct that FedRGD [28] performas aggregation of server and client model in parallel as you described in Fig 4 (a). For FedMatch [27], however, as far as I know, it performs disjoint learning (a bit similar to the alternate training) which trains the global model with labeled data at server “after” local training with unlabeled data is done for the labels-at-server scenario. However, in line 157 to 161, it is not correctly described and so readers may be confused while reading the part.
> > We describe the discrepancy between SemiFL and existing SSFL methods in Section 4 Experiments. ``In particular, FedMatch allocates disjoint model parameters for server and clients, and FedRGD assigns a higher weight for the server model for aggregation. Both methods do not fine-tune the global model with labeled data directly and generate pseudo-labels with the received global model.''

---

> > ### Comment · Reviewer_sMTb · 2022-08-09
> > **Thank you for the response**
> >
> > I thank the authors for providing detailed responses. My curiosity has been addressed. Thank you.

---

> ### Author Response · Authors · 2022-08-06
> **A kind reminder**
>
> Dear Reviewer sMTb,
>
> We would like to thank you again for the time you dedicated to reviewing our paper and your valuable comments. We believe that we have addressed your concerns. Since the end of discussion period is getting close and we have not heard back from you yet, we would appreciate if you kindly let us know of any other concerns you may have, and if we can be of any further assistance in clarifying any other issues.
>
> Thanks a lot again, and with sincerest best wishes
>
> Authors

---

> ### Author Response · Authors · 2022-08-09
> **A kind reminder**
>
> Dear Reviewer sMTb,
>
> We apologize for any inconvenience that our message may cause in advance. Again, we would like to thank you for the time you dedicated to reviewing our paper and your valuable comments. We believe that we have addressed your concerns. Since the end of discussion period is close and we have not heard back from you yet, we would appreciate if you kindly let us know of any concerns you may have, and if we can be of any further assistance in clarifying any other issues. We humbly remain at your disposal.
>
> Thanks a lot again, and with best wishes,
>
> Authors

---

### Official Review · Reviewer_zPN3 · 2022-07-11

**Rating:** 5
**Confidence:** 4
**Soundness:** 3 good
**Presentation:** 3 good
**Contribution:** 3 good

**Summary:**

This work studies an FL setting that clients have completely unlabeled data and can train multiple local epochs, while the server has a small amount of labeled data. For this, author(s) investigate the semi-supervised federated learning scenario and proposes SemiFL method. The focus of the semi-supervised learning (SSL) strategy is o data augmentation-based SSL on a class of nonparametric kernel-based classification learning. The author(s) theoretically methods to the illustrate critical region where significant improvement can be made from unlabeled data. Then, the author(s) propose to augment dataset with pseudo labels by adopting two SSL methods developed in centralized learning: FixMatch and MixMatch. It employs two strategies: fine-tune global model with labeled data and generating labels with global model, to improve the quality of pseudo labeling. Experiments are conducted to validate the proposed method and present clear improvement over the selected baselines.


**Questions:**

Please see the weakness. Plus:
1. The author(s) studied the non-IID data on the client-side. Will the label distribution skewness on the server-side affect SemiFL?
2. How will the theoretical results be generalized to a neural network classifier?
3. Is there any warm-up training to generate the pseudo labels?
4. The reported results on FedMatch seem much lower than those reported in the original FedMatch paper. Could the author(s) please explain the possible reasons?
5. "We use Wide ResNet28x2 [43] as our backbone model for CIFAR10 and SVHN datasets and Wide ResNet28x2 for CIFAR100 datasets throughout our experiments."  What is the reason for mentioning Wide ResNet28x2 twice if all the datasets use the same network architecture?

**Limitations:**

Yes

**Strengths And Weaknesses:**

Strengths:
1.  The paper is easy to follow and the problem is well-motivated.
2.  The proposed method seems very effective, as indicated by the results.

Weakness:
1. It lacks a sufficient connection between the theoretical results and the contribution of the proposed method over the existing FL + SSL methods [27,28].
2. Insufficient hyperparameter study. Although extensive ablation studies are included, it would be helpful to study the confidence threshold in Fixmatch, $a$ in Mixmatch, and $\lambda$ in convex combination on the loss functions.
3. Fair comparison. sBN proposed in HeteroFL [9] is not a technique specifically suitable for the proposed SemiFL. Do the baseline methods also employ sBN?
4. The author(s) mention that the proposed training strategy "delay the computation time." Understanding this limitation without offering the additional computation cost over the total training time is not straightforward.

---

> ### Author Response · Authors · 2022-08-02
> **Response**
>
> Thank you for your time and constructive comments. We have addressed all the comments below. The following major changes will be included in the revision. We hope the responses and planned revisions will be viewed favorably.
>
> 1. It lacks a sufficient connection between the theoretical results and the contribution of the proposed method over the existing FL + SSL methods [27,28].
> > We agree that the paper would be stronger if some convergence results could be derived. However, we found it difficult to develop a comprehensive theory to understand the combination of FixMatch with Federated Learning. In fact, we are not aware of prior theoretical results on the strong data augmentation method used in FixMatch. Therefore, we focus on understanding why and when FixMatch may outperform conventional semi-supervised learning methods. In our original submission, we theoretically analyzed the effect of strong data augmentation on the semi-supervised learning method. In Section C.4 of the appendix (highlighted in the revision), we provided ablation studies of our proposed alternative training by measuring the quality of pseudo-labeling to demonstrate that thresholded pseudo-labels lead to higher accuracy. In summary, we found that 1) thresholded pseudo-labels have much higher accuracy than those without thresholding, and 2) strong data augmentation produces less confident predictions to create a larger loss drop compared with using the originally high confident data.
> 2. Insufficient hyperparameter study. Although extensive ablation studies are included, it would be helpful to study the confidence threshold in Fixmatch, in Mixmatch, and in convex combination on the loss functions.
> > The ablation studies of confidence threshold have been thoroughly discussed in the FixMatch paper, so we did not experimentally study the confidence threshold in our paper. Based on our experiments and ablation studies in the FixMatch paper, our intuition is that the confidence threshold should be close to 1 (e.g., 0.95, 0.99) in order to obtain highly accurate pseudo-labels. However, if the confidence threshold is too large, it may slow down the convergence as the number of labeled data above the confidence threshold (or the labeled ratio) can be insufficient at the beginning of training.
> 3. Fair comparison. sBN proposed in HeteroFL [9] is not a technique specifically suitable for the proposed SemiFL. Do the baseline methods also employ sBN?
> > Yes. To maintain a fair comparison, we have utilized the static Batch Normalization in all of our FL and SSFL methods, including SemiFL, HeteroFl, FedMatch, and FedRGD.
> 4. he author(s) mention that the proposed training strategy "delay the computation time." Understanding this limitation without offering the additional computation cost over the total training time is not straightforward.
> > Our method introduces extra run time costs because we need to fine-tune the averaged global model with the labeled data at the server at each communication round. In our experiments, the clients and server are trained with the same number of epochs.
> 5. The author(s) studied the non-IID data on the client-side. Will the label distribution skewness on the server-side affect SemiFL?
> > If the labels at the server are imbalanced, one may adopt techniques such as oversampling during fine-tuning to address such problems. The accuracy of labels at the server is the key to correcting the errors accumulated from training unlabeled data at clients.
> 6. How will the theoretical results be generalized to a neural network classifier?
> > The current theoretical analysis needs an explicit form of how a single observation may influence the trained classifier (through a smoothing kernel formula). For neural networks, however, we do not know how to explicitly connect the estimated parameters (neural weights) with observations.
> 7. Is there any warm-up training to generate the pseudo labels?
> > The proposed method does not need warm-up training. Our intuition is that the model will be roughly trained with labeled data and a small amount of high-confidence pseudo-labeled data at the beginning. As the round goes, the model will be more accurate, generating more high-confidence data and boosting the effective training size.
> 8. The reported results on FedMatch seem much lower than those reported in the original FedMatch paper. Could the author(s) please explain the possible reasons?
> > The original FedMatch paper uses 5000 labeled data with ResNet9 for the CIFAR10 dataset, but our paper uses 4000 labeled data with WResNet28x2 to compare with the SSL method. It can be verified that training 5000 labeled data with ResNet9 alone can obtain more than 75\% accuracy, but FedMatch only reports 52\% (Table 1), which indicates that FedMatch does not improve the performance of the labeled server.

---

> > ### Author Response · Authors · 2022-08-02
> > **Response continue**
> >
> > "We use Wide ResNet28x2 [43] as our backbone model for CIFAR10 and SVHN datasets and Wide ResNet28x2 for CIFAR100 datasets throughout our experiments." What is the reason for mentioning Wide ResNet28x2 twice if all the datasets use the same network architecture?
> > > This was a typo. We have corrected the second part of the sentence to ``Wide ResNet28x8 for CIFAR100 datasets'' (highlighted in revision). We use ResNet28x8 for CIFAR100 for a fair comparison with the results reported in SSL papers.

---

> > > ### Comment · Reviewer_zPN3 · 2022-08-08
> > > **Thanks for your response**
> > >
> > > Thanks for the authors' replies. Some of my questions and concerns are addressed, but I still have some questions and concerns.
> > >
> > > 1. I appreciate that the authors studied threshold and pseudo-label quality. Could the authors elaborate more on the 'critical region' of $n_u$ in the main theorem? For the assumptions, the authors introduced a few properties for augmentation. Can the authors validate or explain how their augmentation methods can meet the assumptions or are motivated by the theoretical understanding? Finally, it seems the theoretical results are not specific to FL, right?
> > > 2. The authors' replies list many differences between the proposed methods vs. FedMatch. I am not convinced that  'The ablation studies of confidence threshold have been thoroughly discussed in the FixMatch paper, so we did not experimentally study the confidence threshold in our paper.' is a good reason to omit hyper-parameter discussion for the proposed method.
> > > 3. It is still unclear about the effect of "delay the computation time" without quantitative measurement.
> > >
> > > Minor note: the link to your revision is broken.

---

> > > > ### Author Response · Authors · 2022-08-08
> > > > **Response**
> > > >
> > > > Thanks for your constructive comments.
> > > > 1. $n_u$ indicates the amount of unlabeled data that we need to make significant improvement from using only labeled data. Intuitively, strong augmentation is
> > > > a process that maps a data point (e.g., an image) from high quality to relatively low quality. In particular, the predictions of augmentated samples are still acurrate (argmax) but less confident. The theoretical results are not for FL but for SSL in general. We could not find a therotical analysis for strong data augmentation for SSL. In order to better understand the effectiveness of strong data augmentation for SSL, we develop the therotical analysis inspired by many experiments we conducted.
> > > > 2. FedMatch also utilizes the same confidence threshold as in FixMatch. We use the value 0.95 suggested by the FixMatch paper for our experiments. We provided some intuition on the confidence threshold in our previous response. Due to the time constraint, we will add the ablation studies of confidence threshold in our next revision.
> > > > 3. The computation time can be measured by the number of training epochs of the server and clients. By assuming the server and clients have the same computational power, the server and clients will consume the same amount time. In practice, the server typically has more computation resources, and thus the delay of the computation time may not be significant.

---

> ### Author Response · Authors · 2022-08-06
> **A kind reminder**
>
> Dear Reviewer zPN3,
>
> We would like to thank you again for the time you dedicated to reviewing our paper and your valuable comments. We believe that we have addressed your concerns. Since the end of discussion period is getting close and we have not heard back from you yet, we would appreciate if you kindly let us know of any other concerns you may have, and if we can be of any further assistance in clarifying any other issues.
>
> Thanks a lot again, and with sincerest best wishes
>
> Authors

---

### Official Review · Reviewer_rcxY · 2022-07-12

**Rating:** 3
**Confidence:** 4
**Soundness:** 2 fair
**Presentation:** 2 fair
**Contribution:** 2 fair

**Summary:**

-This paper studies the semi-supervised federated learning problem, where the server has access to a limited number of labeled data while the clients have access to massive unlabeled data. They claim that directly combining existing semi-supervised learning methods such as FixMatch and MixMatch with standard federated learning methods such as FedAvg cannot perform well, since the server and clients models are trained in parallel and pseudo labels for the unlabeled data are generated by the trained models for every batch, and they theoretically show that its performance depends on the quality of the augmented pseudo-labeled data: the higher quality, the better risk bound. Then, they propose to train the labeled server and the unlabeled clients in an alternate manner: they generate pseudo-labels only once upon receiving the server model and fine-tune the server model by additional training with labeled data. They also provide some favorable experimental results in the paper.


**Questions:**

-In the definition of adequate transmission (line 140-144), how to quantify the reliable information exhibited from unlabeled data, can any information-theoretic measures be used here? Otherwise, the definition looks very unclear to me.

-In Figure 1, it seems that the unlabeled clients can use different models (illustrated by smartphone vs tablet vs laptop…), which may cause some misunderstandings since in line 106 it is assumed that all models share the same architecture.

-In Table 2, it seems that fine-tuning the server model with labeled data helps improve performance the most (almost 40% improvement). But in practice, the number of labeled data may vary. Any results or insights on how many labeled data are sufficient for the fine-tuning?

-The proposed method is expected to improve the quality of generated pseudo-labels during training in an alternating training fashion. It would be more convincing if the results of the generated pseudo-labels during training can be provided.

-The proposed method introduces several hyperparameters, for example, the global confidence threshold tao which needs to be preselected by the clients. This may be difficult since the clients only receive the global model from the server and may not have other prior knowledge, how to select tao in practice?


**Limitations:**

-The paper has many unclear parts and needs to be carefully modified.

**Strengths And Weaknesses:**

-------------------------------
pros:

-The paper considers a practical federated learning problem and is motivated by relevant real-world examples: in a healthcare system, a central hub (server) has domain experts and a limited number of labeled data, while many rural branches have non-experts and massive unlabeled data. It would be more convincing if the effectiveness of the proposed method can also be verified on those real-world datasets.

-------------------------------
cons:

-The writing of the paper may not be clear, and many confusing parts need to be modified. For example, in lines 38-39, it is said that it is **infeasible** to directly combine the federated learning method (FedAvg) with semi-supervised learning methods. This statement may be too strong, as discussed in lines 158-159 some methods e.g., FedMatch can do this.

-Also in lines 131-133, it is said that a successful semi-supervised learning method **must** be able to generate more and more high-quality pseudo-labels during training…This statement may also be too strong as in standard semi-supervised learning, the unlabeled data are used for regularization and the good generalization is based on the cluster assumption, the manifold assumption…

see section 1.2 in the following textbook:

Chapelle, Olivier, Bernhard Schölkopf, and Alexander Zien. Semi-Supervised Learning. MIT Press.

Such a statement may only hold for the pseudo-label generation-based semi-supervised learning approaches.

-Some abbreviations firstly appear without the full words and citations, see lines 114-116, FixMatch, FedSGD, MixMatch…

-**Lack of convergence results**. The proposed method is based on alternating training, which may be unstable in some situations. It would help make the paper stronger if some convergence results of the proposed algorithm can be given.

---

> ### Author Response · Authors · 2022-08-02
> **Response**
>
> Thank you for your time and constructive comments. We have addressed all the comments below. The following major changes will be included in the revision. We hope the responses and planned revisions will be viewed favorably.
> 1. The writing of the paper may not be clear, and many confusing parts need to be modified. For example, in lines 38-39, it is said that it is infeasible to directly combine the federated learning method (FedAvg) with semi-supervised learning methods. This statement may be too strong, as discussed in lines 158-159 some methods e.g., FedMatch can do this.
> > Following your comment, we have rephrased the statement to ``It is challenging to directly combine semi-supervised learning methods with the communication efficient federated learning method like FedAvg to allow local clients to train multiple epochs.''
> 2. Also in lines 131-133, it is said that a successful semi-supervised learning method must be able to generate more and more high-quality pseudo-labels during training… This statement may also be too strong as in standard semi-supervised learning, the unlabeled data are used for regularization and the good generalization is based on the cluster assumption, the manifold assumption…
> > We have rephrased the statement to: ``The main idea is to construct a pseudo-labeling mechanism whereby our SSL method can generate more and more high-quality pseudo-labels during training. Meanwhile, the augmented data for model training can produce a considerable drop in training loss than compared with the original data.''
> 3. Lack of convergence results. The proposed method is based on alternating training, which may be unstable in some situations. It would help make the paper stronger if some convergence results of the proposed algorithm can be given.
> > We agree that the paper would be stronger if some convergence results could be derived. However, we found it difficult to develop a comprehensive theory to understand the combination of FixMatch with Federated Learning. In fact, we are not aware of prior theoretical results on the strong data augmentation method used in FixMatch. Therefore, we focus on understanding why and when FixMatch may outperform conventional semi-supervised learning methods. In our original submission, we theoretically analyzed the effect of strong data augmentation on the semi-supervised learning method. In Section C.4 of the appendix (highlighted in the revision), we provided ablation studies of our proposed alternative training by measuring the quality of pseudo-labeling to demonstrate that thresholded pseudo-labels lead to higher accuracy. In summary, we found that 1) thresholded pseudo-labels have much higher accuracy than those without thresholding, and 2) strong data augmentation produces less confident predictions to create a larger loss drop compared with using the originally high confident data.
> 4. In the definition of adequate transmission (line 140-144), how to quantify the reliable information exhibited from unlabeled data, can any information-theoretic measures be used here? Otherwise, the definition looks very unclear to me.
> > In the notion of adequate transmission, the reliable information exhibited from unlabeled data is reflected in two aspects. First, if conditioning on both input signals (e.g., images), the output label has a distribution that is only determined by the higher-quality input, which is quite intuitive (Condition A2 in Appendix D.3). Second, for every regime of significant interest in evaluating the prediction performance, there will be a sufficient probability coverage of the augmented data (Condition A3 in Appendix D.3). It ensures the augmented data can represent the test data of interest to boost the test performance.
> 5. In Figure 1, it seems that the unlabeled clients can use different models (illustrated by smartphone vs tablet vs laptop…), which may cause some misunderstandings since in line 106 it is assumed that all models share the same architecture.
> > We have revised the figure as you suggested.
> 6. In Table 2, it seems that fine-tuning the server model with labeled data helps improve performance the most (almost 40\% improvement). But in practice, the number of labeled data may vary. Any results or insights on how many labeled data are sufficient for the fine-tuning
> > We conduct ablation studies of the number labeled data in Table 1. For example, we experimented with 250 and 4000 labeled data for the CIFAR10 dataset. Compared with  FixMatch trained on centralized data, SemiFL has more performance drop due to an insufficient amount of labeled data. However, we believe it is foreseeable since the problem scenario restricts us from sampling labeled and unlabeled data in one batch.

---

> > ### Author Response · Authors · 2022-08-02
> > **Response continue**
> >
> > 7. The proposed method is expected to improve the quality of generated pseudo-labels during training in an alternating training fashion. It would be more convincing if the results of the generated pseudo-labels during training can be provided.
> > > We have added the results of generated pseudo-labels during training in Table 2 and Appendix C.4. Our results demonstrate that `Fine-tuning with labeled data` can produce results better than the `Partially Supervised`case, and `Generate pseudo-labels with global model` further improves the performance to be close to the centralized SSL method.
> > 8. The proposed method introduces several hyperparameters, for example, the global confidence threshold tao which needs to be preselected by the clients. This may be difficult since the clients only receive the global model from the server and may not have other prior knowledge, how to select tao in practice?
> > > We assume that the confidence threshold $\tau$ can be preselected by the server. In practice, it can be selected using validation data. Based on our experiments and ablation studies in the FixMatch paper, our intuition is that the confidence threshold should be close to 1 (e.g., 0.95, 0.99) to obtain highly accurate pseudo-labels. However, if the confidence threshold is too large, it will slow down the convergence as the number of labeled data above the confidence threshold (or the labeled ratio) may be insufficient at the beginning of training.

---

> ### Author Response · Authors · 2022-08-06
> **A kind reminder**
>
> Dear Reviewer rcxY,
>
> We would like to thank you again for the time you dedicated to reviewing our paper and your valuable comments. We believe that we have addressed your concerns. Since the end of discussion period is getting close and we have not heard back from you yet, we would appreciate if you kindly let us know of any other concerns you may have, and if we can be of any further assistance in clarifying any other issues.
>
> Thanks a lot again, and with sincerest best wishes
>
> Authors

---

> ### Author Response · Authors · 2022-08-09
> **A kind reminder**
>
> Dear Reviewer rcxY,
>
> We apologize for any inconvenience that our message may cause in advance. Again, we would like to thank you for the time you dedicated to reviewing our paper and your valuable comments. We believe that we have addressed your concerns. Since the end of discussion period is close and we have not heard back from you yet, we would appreciate if you kindly let us know of any concerns you may have, and if we can be of any further assistance in clarifying any other issues. We humbly remain at your disposal.
>
> Thanks a lot again, and with best wishes,
>
> Authors

---

### Meta-Review · Area_Chair_9buH · 2022-08-23

**Recommendation:** Accept
**Confidence:** Less certain

**Metareview:**

There was some disagreement in reviewer scores for this paper, which proposes an algorithm for semi-supervised federated learning. The recurring concerns centered on a few points: (i) the lack of theoretical analysis of the proposed algorithm, (ii) the apparent disconnect of the presented analysis from the setting under consideration, (iii) limited technical novelty, (iv) potentially missing baselines, and (v) lack of clarity in the text. Of these, the response to (i), (iii) and (v) seem reasonable: certainly a systematic analysis of a new algorithm is welcome, but hardly straightforward; simple ideas are to be commended if they definitively solve a challenging problem; and the proposed edits do go some way to addressing the reviewer concerns.

Points (ii) and (iv) remain less clear. For the former, it is a bit confusing to present a generic analysis of SSL techniques, which does not appear to directly influence the proposed algorithm design. For the latter, there are some missing references at a minimum, e.g., Zhang et al., "Benchmarking Semi-supervised Federated Learning"; Itahara et al., "Distillation-Based Semi-Supervised Federated Learning for Communication-Efficient Collaborative Training with Non-IID Private Data". Further, some of the results in Table 1 do not seem to align with what has been previously reported in the literature; e.g., Table 1 of the FedMatch paper. However, it appears that the authors have at least compared against the most recent Fed-SSL techniques, with unsupervised and self-supervised techniques being not directly in scope.

Overall the paper is a borderline case. The recurring critique of limited depth suggests that a compelling empirical analysis would be appropriate. The explicit evaluation of Fed-SSL methods against a labelled-only baseline, and showing existing methods cannot outperform this, is interesting. The paper could be strengthened by compressing Sec 3.2 and focussing instead on a better understanding of why existing Fed-SSL methods underperform. Such insights would be more broadly insightful to the community, and be a useful contribution.

**Award:**

No

---

### Decision · Program_Chairs · 2022-09-14

Accept